

# Data efficient Random Forest model for avalanche forecasting

Manesh Chawla[1], Amreek Singh[2]

[1]Snow and Avalanche Study Establishment,  Manali - 175103, India
[2]Snow and Avalanche Study Establishment,  Chandigarh - 160037, India

*Correspondence to*: Manesh Chawla (zmfzmj123@gmail.com)

**Abstract.**

Fast downslope release of snow (avalanche) is a serious hazard to people living in snow bound mountains. Released snow mass can gain sufficient momentum on its down slope path to kill humans, uproot trees and rocks, destroy buildings. Direct
reduction of avalanche threat is done by building control structures to add mechanical support to snowpack and reduce or deflect downward avalanche flow. On large terrains it is economically infeasible to use these methods on each high risk site. Therefore predicting and avoiding avalanches is the only feasible method to reduce threat but sufficient snow stability data for accurate forecasting is generally unavailable and difficult to collect. Forecasters infer snow stability from their knowledge of local weather, terrain and sparsely available snowpack observations. This inference process is vulnerable to
human bias therefore machine learning models are used to find patterns from past data and generate helpful outputs to minimise and quantify uncertainty in forecasting process. These machine learning techniques require long past records of avalanches which are difficult to obtain. In this paper we propose a data efficient Random Forest model to address this problem. The model can generate a descriptive forecast showing reasoning and patterns which are difficult to observe manually. Our model advances the field by being inexpensive and convenient for operational forecasting due to its data
efficiency, ease of automation and ability to describe its decisions.

## 1 Introduction

In snow bound areas avalanches cause loss of life and property worldwide. Avalanche deaths are estimated at 250 per year (Schweizer et al., 2015). Government and private agencies are funded to reduce avalanche threat for important activities and property e.g road/rail transport, construction, army movement etc. This effort and research funding has led to development of
several techniques to reduce avalanche threat. For a specific site [< 1 km$^2$ ], avalanche threat is reduced by building control structures, modification of nearby terrain or use of explosives to trigger avalanches in controlled way (Fuchs et al., 2007). Using such techniques on each risk site over large areas is economically infeasible therefore avalanche forecasting is done to plan passive risk reduction measures.  Individuals can use information in forecast to plan their activities in snow bound areas.





Avalanche forecasting aims to identify the location of snowpack weakness and its triggering risk. Observing snowpack stability at a high spatio temporal resolution over large terrain is a difficult problem. Therefore stability at most risk sites is deduced using secondary observable data e.g meteorological and snowpack parameters from a similar representative site, terrain parameters of the site, expected changes to snowpack by weather etc. Snow stability shows high variance with respect
to terrain features. Deduction process for snow stability from secondary data has not been mathematically formulated therefore forecasters need to rely on their intuition of local terrain and snowpack patterns to estimate stability and collect more information to minimise uncertainty( LaChapelle, 1980, Schweizer et al., 2008 ). Numerical and statistical models are important tools for adding objectivity to this process.


Numerical models simulate the snowpack and weather processes that contribute significantly to avalanche hazard. CROCUS and SNOWPACK give accurate snow profile simulations at a microscale level( $< 1$ km$^2$ ) for sites where meteorological data is available (Vionnet et al., 2012; Lehning et al., 1999). Meteorological sensors cannot be setup at all risk sites therefore interpolated meteorological data from numerical weather models like SAFRAN is used as input for SNOWPACK model
(Lehning et al., 1999). The output from SNOWPACK tells forecasters about slopes where snowpack stability is changing due to numerically modelled snowpack processes : weak layer formation due to temperature gradients, surface or deep wetting, compaction and refreezing (Lehning et al., 1999). A limitation of this model chain is its inability to account for some contributory processes e.g wind loading, its accuracy can be seriously affected by errors in interpolated meteorological data.

Statistical models take input from a specific site and use it as representative of conditions over a larger region (mesoscale ~ 10 km$^2$) to estimate the avalanche threat. These models link weather and snowpack variables to avalanche threat using avalanche occurrences from historical data (Buser., 2009; Gassner et al., 2001). Information from multiple sources( possibly redundant) e.g wind loading indexes, local terrain features, location specific snowfall patterns , numerical snowpack simulations can be included in these models (Pozdnoukhov et al., 2011) , this makes them more robust to errors in individual
parameters compared to numerical models. Forecasts of numerical models can be improved by using their results in statistical models.

Machine learning has been used for tasks where procedures cannot be precisely formulated but humans perform well e.g in handwriting and speech recognition (Liang and Hu, 2015). Machine learning models are not used to automate avalanche
forecasting process, they instead help forecasters judgement by providing information from past data relevant to the forecasted day. In this paper we build a machine learning model using random forest technique. The model gives interpretable data mining outputs and is convenient to use for operational applications due to its data efficiency and automation.

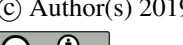




Nearest neighbours model is a frequently used statistical model for avalanche forecasting (Buser., 2009; Singh and Ganju, 2008; Gassner et al., 2001; Singh et al., 2014) . It estimates threat by using a set of historical days most similar to the forecasted day. It is unable to directly model the inductive reasoning process used by avalanche forecasters, this may cause

data inefficiency. Models in (Buser, 2009; Gassner et al., 2001; Singh and Ganju, 2008; Singh et al., 2014) required atleast 7 years training data.

Decision trees and expert systems have been used to model complex patterns which are missed by nearest neighbours (Rosenthal et al., 2001; Singh and Ganju, 2008; Schweizer and Föhn, 1996; Heindrikx et al. 2014) . These techniques are

capable of using expert knowledge by modeling known forecasting rules. Unfortunately individual trees are sensitive to small changes in data and unable to learn complex decision boundaries without overfitting (Hastie et al., 2009). Expert systems can be designed to satisfy all the criterion mentioned above but considerable human effort and expertise is required to build expert systems.

(Pozdnoukhov et al., 2008) use support vector machines (SVM) on high dimensional feature vectors for avalanche forecasting. They use feature vectors from multiple data sources for geo spatial forecasting (Pozdnoukhov et al., 2011), these vectors include several features representing slope ,elevation, snow drift, snow stability and meteorological parameters. SVM maybe difficult to interpret by a forecaster, in (Pozdnoukhov et al., 2011) it is proposed to explore the support vectors for interpreting model outputs, some features used by authors implementation currently require manual effort to record.


A model satisfying following criterion can be made operational at a low cost :

1. Can be trained to give acceptable performance using a low amount of historical data. This makes it useful for regions where long and reliable historical records are unavailable.

2. Can forecast using only data collected from automated sensors. Data with high spatio-temporal resolution can be used from a grid of sensors.

3. Can explain the reasoning used to arrive at conclusion and gives numerical estimates justifying the reasoning. The explaination should not require significant forecasting experience to interpret.

4. Acceptable forecast skill scores for operational use, high risk days should be detected with low rate of false positives.


We use an ensemble learning technique (Random Forest) , an ensemble of decision trees gives the  prediction. Random forest ensemble can learn complex decision boundaries and is resistant to overfitting (Briemann., 2001). Decision trees have an



interpretable output, trees from ensemble have been used to build a descriptive forecast. Our model satisfies all four criterion

above, therefore it overcomes some of the problems caused in operational use of the models we surveyed. The model can be

deployed at sites where data of three winters is available. In future we will explore the use of transfer learning techniques to

reduce the data requirement further.

## 2 Modeling Technique

### 2.1 Random Forest

Random forest is an ensemble learning method (Opitz and Maclin, 1999). Individual decision trees have weak performance

due to overfitting and high variance. Random forest uses a collection of decision trees to improve prediction. Each tree is

trained on a random dataset derived from the training data using a process called bagging, this ensures that individual trees

are uncorrelated (Brieman, 1996). The output of the collection for a data point is given by the mean output of trees at that

point. The ensemble model is partially interpretable and depends on few parameters. Some useful properties of model are

(Briaman and Friedman, 1984):

        1. A method for ranking feature importance.

        2. Robust to outliers and missing values.

115        3. Can handle both discrete and continuous features without special pre-processing.

        4. Training process can be highly parallelized by training trees on separate threads.

        5. State of the art accuracy on various tasks (Rogez et al., 2008).

### 2.2 Decision Trees

A decision tree describes a flow chart like process for classifying data points. Each non leaf node in the tree defines a test on

the data point, each leaf node defines a classification. To classify a point we apply the test at root node to a data point,

depending on the result we move to a child node. If child node is a non-leaf node the same process is repeated to move to

subsequent child node. This is repeated till a leaf node is reached, this leaf node defines the classification for the data point.

We demonstrate this through an example (cf Figure 1).



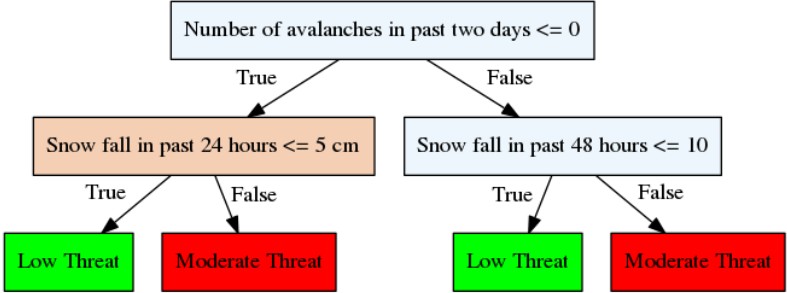

**Figure 1: An example tree classifying a day as moderate or low avalanche threat using avalanche occurences and snowfall data of past 2 days.**

If for some day the following parameters are known to be:

    1. Snowfall in past 24 hours = 48 cm

    2. Snowfall in past 48 hours = 80 cm

    3. Number of avalanches in past 2 days = 0

The test on root is logically equivalent to testing if there were any avalanches in past two days, since there were no avalanches in past two days we move to left child node as directed by arrows. We apply the test in left child node which returns false since snowfall was 48 cm in past 24 hours, therefore we move to right child node which classifies the day as moderate risk.

**2.3 Training Decision Trees**

Training algorithms for decision trees proceed by splitting the training dataset based on a feature value such that the resulting datasets are more homogeneous in their target variable, this splitting process continues recursively on split datasets till a termination criteria is reached which specifies that the dataset is sufficiently homogenous. Recursive splitting process naturally defines the decision tree. Each node corresponds to a dataset and the split mentioned in node corresponds to the split decided by the training process.


Algorithm starts with splitting the entire training dataset, writes the split in root node and adds two child nodes (left child C1 and right child C2) to root. Split datasets correspond to the child nodes and are split further adding more child nodes to the nodes C1 and C2. This process can therefore be repeated recursively till sufficiently homogenous datasets are formed which
are represented by leaf nodes. The majority classification label of points in these final homogenous datasets is taken as the label represented by the corresponding leaf node. See code in Figure 2 for details.



```
train(node)
{
        if( stopping criteria at node is satisfied )
        {
                //exits the function call
                return;
        }

        else
        {
                //Select the splitting feature and store it in split_feature variable
                split_feature = select_split_feature(node);

                //Compute the splitting threshold for the selected feature in node
                split_threshold = compute_split_threshold(node, split_feature);

                //Initialise and append the left and right child nodes to parent node
                left_child = Child node initialised with all data points in parent node with split_feature < split_threshold

                right_child = Child node initialised with all data points in parent node with split_feature >= split_threshold

                //Grow the tree recursively
                train(left_child);
                train(right_child);

        }
}|
```

**Figure 2: Pseudocode for decision tree training.**

In this paper we use C4.5 algorithm implemented in Scikit-learn, a python machine learning library (Bressert, 2012; Quinlan, 1993) . C4.5 splits on the attribute with highest normalised information gain. In later sections we use Gini coefficient to measure the homogeneity of target variables. Gini coefficient of observations $y_1, y_2, ...., y_n$ is defined by:

$$G(y_1, y_2, ....., y_n) = \frac{\sum\sum |y_i - y_j|}{\sum\sum y_j} = \frac{\sum\sum |y_i - y_j|}{n\sum y_j} \qquad \text{....... Eq (1)}$$

If all values are almost equal G approaches 0, if few values dominate all others G approaches 1.

**2.4 Random Forest Training**

Trees are trained on subspaces of the dataset. A subspace is formed by drawing a random sample with replacement from training set and then selecting a random subset of features from the drawn sample. To build the ensemble a user specified number of decision trees are trained and stored in memory, each tree is trained on a independent random subspace of the training data.






## 3 Features and Dataset

The model has been trained and tested using snow-meteorological and avalanche occurrence observations from Bandipore-Gurez (BG) sector in the state of Jammu & Kashmir of India (Figure 4). BG road axis mainly runs along the Kishenganga river in Gurez and Tulel valleys at the tip of Great Himalayan Range in northwest Himalaya and connects to Bandipore town in Kashmir valley through Razdaan pass (3300m above msl). In Gurez valley (area on west of Wampore town), 40 major avalanche sites affect the highway stretch of about 25 kms from Jatkushu village to Wampore village. Besides, about 15 avalanche sites affect the lateral tracks. In Tulel valley (area on east of Wampore town), over 100 major and minor avalanche sites affect the highway and lateral tracks. The formation zone altitude of avalanche sites in the area ranges from about 2350m to 4800m above msl. A snow-meteorological observatory is located near Kanzalwan. The area is characterized by continental climate and receives moderate to heavy snowfall during winter season.

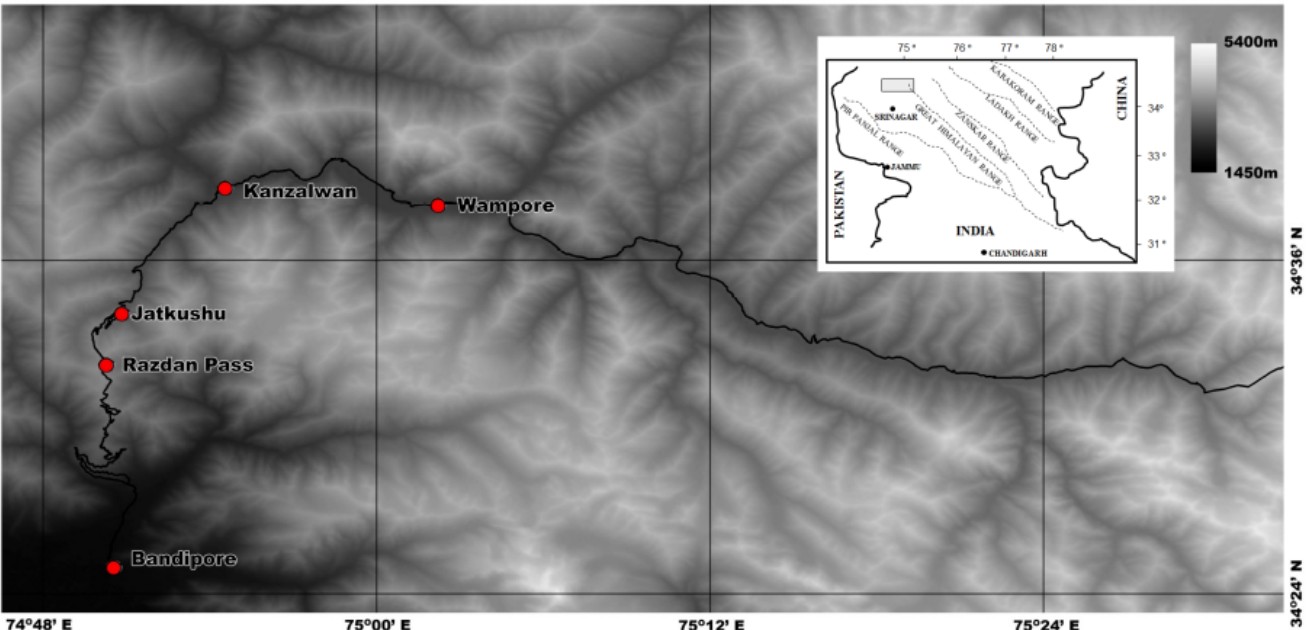

**Figure 3: A perspective view of BG sector.**

The set of input parameters used for forecasting risk on a day are summarized in Table 1 and Table 2. While parameters in Table 1 can be observed in automated mode, parameters in Table 2 are derived from these along with avalanche occurrence data to represent the events of past few days. The prediction for a day can be done automatically if avalanche occurences of past days are known, these can be detected automatically using infrasonic sensors and radars.




**Table 1: Parameters used for prediction. These can be automatically recorded.**

| Parameter name | Unit | Description |
|---|---|---|
| MAX-TEMP | Degree Celcius ( °C ) | Maximum Temperature of past 24 hours |
| MIN-TEMP | Degree Celcius ( °C ) | Minimum Temperature of past 24 hours |
| SNOW-TEMP | Degree Celcius ( °C ) | Snow surface temperature measured at 0830 hours |
| SNOW-DEPTH | Meter ( m ) | Height of snow surface above ground level |
| SNOW-AMT | Meter ( m ) | Amount of snowfall in past 24 hours |
| AVG-WIND-SPEED | Meter/Second ( m/s$^{-1}$ ) | Average wind speed in past 24 hours |


**Table 2: Parameters derived from observed parameters. *Represents parameters derived using training labels.**

| Parameter name | Unit | Description |
|---|---|---|
| SNOW-TEMP-DIFF | Degree Celcius ( °C ) | Snow surface temperature difference from past day. |
| SNOW-AMT-2 | Meter (m) | Snow fall in past 2 days. |
| SNOW-AMT-4 | Meter (m) | Snow fall in past 4 days. |
| SNOW-AMT-10 | Meter (m) | Snow fall in past 10 days. |
| NUM-AVALANCHES-2* | None | Number of avalanches in past 2 days |
| NUM-AVALANCHES-4* | None | Number of avalanches in past 4 days |
| AVG-WIND-SPEED-2 | Meter/Second ( m/s$^{-1}$ ) | Average wind speed of past 2 days |
| AVG-WIND-SPEED-4 | Meter/Second ( m/s$^{-1}$ ) | Average wind speed of past 4 days |
| AVG-WIND-SPEED-10 | Meter/Second ( m/s$^{-1}$ ) | Average wind speed of past 10 days |






## 4 Model performance and parameters

We define a confusion matrix of a classifier C as on a labeled dataset D as:

$$a_{ij} = |S_{ij}|$$

Where $S_{ij}$ is defined as:

$$S_{ij} = \{x \in D: \quad C(x) = i \quad \text{and} \quad label(x) = j\} \qquad \text{Eq (2)}$$

This matrix is used for performance analysis of classifiers. Here we derive the following measures from confusion matrix to describe performance.

**Table 3: Performance measures used for model validation.**

| Measure name | Description | Expression in terms of confusion matrix |
|---|---|---|
| False Alarm Rate(FAR) | Conditional probability of returning an avalanche day given underlying day is non avalanche day. | $\dfrac{a_{10}}{\sum a_{i0}}$ |
| Probability of Detection(POD) | Conditional probability of forecasting an avalanche day given underlying day is avalanche day. | $\dfrac{a_{11}}{\sum a_{i1}}$ |
| Precision | Fraction of predicted days which are avalanche days. | $\dfrac{a_{11}}{a_{10} + a_{11}}$ |
| Heidke Skill Score (HSS) | Measures the forecast performance of classifier over of a defined random forecast (Wilks, 1995). | $\dfrac{a_{11} a_{00} - a_{10} a_{01}}{(a_{11} + a_{01})(a_{01} + a_{00}) + (a_{11} + a_{10})(a_{10} + a_{00})}$ |
| Hansen Kuipers Skill Score (TSS) | Measures the forecast performance of classifier over of a defined random forecast (Wilks, 1995). | $G(y_1, y_2, \ldots, y_n) = \dfrac{\sum \sum |y_i - y_j|}{\sum \sum y_i} = \dfrac{\sum \sum |y_i - y_j|}{n \sum y_i}$ |




### 4.1 Data Preprocessing

Model was trained on winter data from December 2010 to March 2013(Three winter seasons ), validation was done on
winter data from December 2013 to March 2017. The dataset contains more non avalanche examples than avalanche days
[Table 4]. Training a classifier on this dataset will bias it to forecast more non avalanche days. A solution is to use cost
corrected classifiers with higher cost assigned to minority examples. Another approach is to discard majority class data
randomly or synthetically generate more minority class data to make class sizes equal. This approach can lead to overfitted
classifier when datasets are highly skewed.


To reduce skewness we remove from training and testing dataset all days for which avalanches are unlikely due to lack of
sufficient standing snow. We discard examples where snow height is less than 50cm [Table 4]. This filtering step removes
poor examples which cause overfitting of trees. See Table 4 for justification of threshold choice and summary statistics of the
dataset. When training decision trees of ensemble, the classes are weighted by their proportion in filtered dataset.


**Table 4: Summary statistics of BG axis avalanche dataset ( Dec 2010 – Mar 2017)**

| | |
|---|---|
| Number of days in winter season | 120 |
| Mean number of avalanches per season | 30 |
| Mean number of avalanches per season when snow height was greater than 50 cm. | 29 |
| Mean number of days out of 120 days per season when snow height is greater than 50 cm. | 81 |
| Mean number of non-avalanche days when snow height was greater than 50 cm. | 52 |



## 4.2 Parameter Tuning

We tune the following model parameters:

D = Maximum depth allowed for a tree added to ensemble.

 N = Number of trees used in ensemble.

Model output is the estimated probability of an avalanche given the input parameters for the day defined in Table 1. To get a

binary classification and validate the model using scores in Table 3 a threshold is selected. If risk prediction for a day is greater than the selected threshold it is classified as avalanche day otherwise a non avalanche day. Threshold choice sets a trade off between the risk of missing an avalanche day against the risk of false alarm on a non avalanche day. Low threshold values give high false alarm rates but fewer avalanche days are misclassified as non avalanche days. High threshold choice gives few false alarms but misses more avalanche days. Forecaster can select a threshold optimal for his risk management

strategy without retraining the model. To evaluate model performance for a set of parameters (N,D) we use HSS scores obtained against possible threshold values.

Increasing values of D beyond 3 decreased the  model performance as measured by HSS on most threshold choices. Increasing D leads to overfitting of individual trees and lowers their performance. This leads to lowered performance of

ensemble.

 Increasing N reduces the variance in model prediction, and smooths decision boundaries. Increasing N beyond the value required for convergence of  model gives no significant change in risk outputs and performance. Values of N beyond 5000 had no significant impact on model performance metrics for values of D between 2 and 5. For higher values of D model

performance was unacceptable for values of N tested (2000 <= N <= 100000).

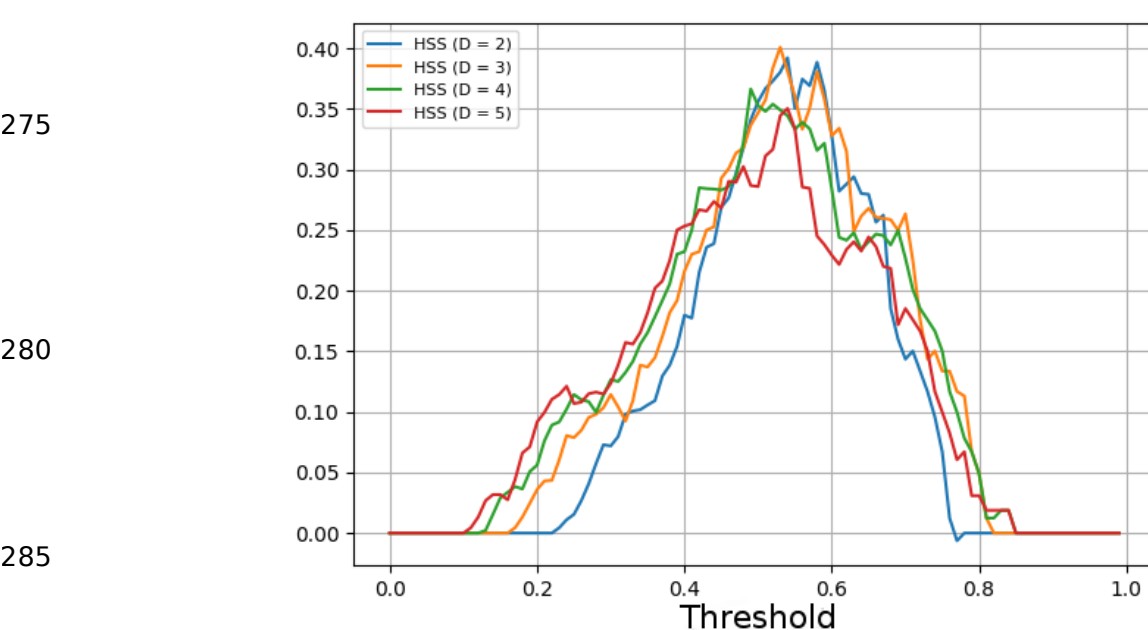

**Figure 4: HSS at various thresholds used for parameter tuning of Random Forest model. Scores decrease for values of D beyond 3.**

**4.3 Model Validation**

Figures 6,7 show model performance using FAR, POD and Precision scores. Precision scores show increasing trend with threshold. Beyond threshold of 0.82 precision is set zero, this is because 0.82 is the highest predicted probability on the testing data. This shows the difficulty of avalanche forecasting, due to unpredictable causes of avalanches classifier cannot be confident above the threshold.




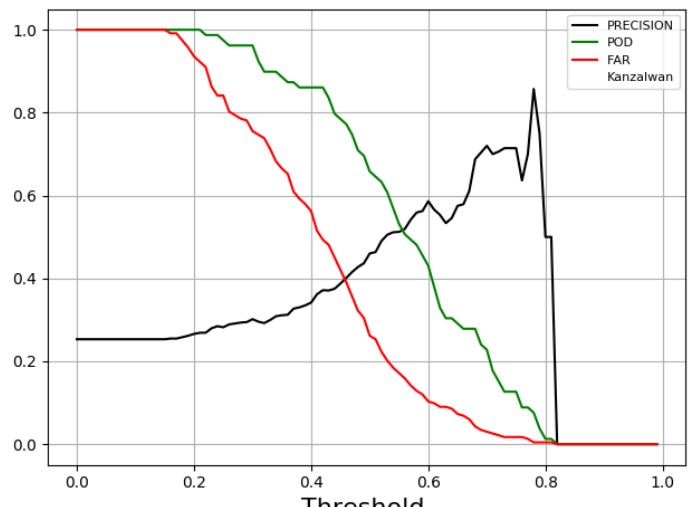

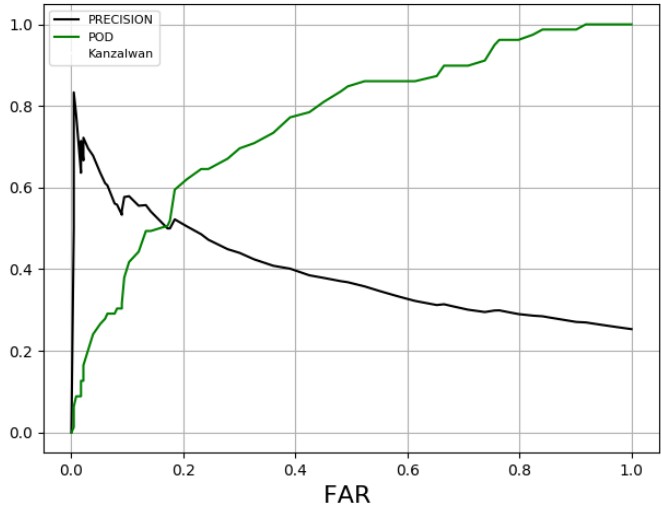

**Figure 5: Graph showing FAR,POD and Precision scores at various threshold choices.**

**Figure 6: POD and Precision against FAR values tolerated.**

**Table 5 : POD and Precision values for different FAR tolerated when using Random Forest model for forecasting.**

| FAR | POD | PRECISION |
|---|---|---|
| 0.1 | 0.43 | 0.59 |
| 0.15 | 0.5 | 0.52 |
| 0.2 | 0.62 | 0.51 |
| 0.25 | 0.66 | 0.47 |
| 0.3 | 0.7 | 0.44 |
| 0.35 | 0.74 | 0.41 |
| 0.4 | 0.77 | 0.39 |
| 0.45 | 0.82 | 0.38 |
| 0.5 | 0.86 | 0.37 |
| 0.55 | 0.86 | 0.35 |
| 0.6 | 0.87 | 0.33 |
| 0.65 | 0.87 | 0.31 |
| 0.7 | 0.9 | 0.30 |




For high classification threshold only days when the model is confident are classified as positive. Increasing threshold improves precision and lowers false alarms. This shows that the model estimates avalanche risk consistently, days predicted with high confidence are more likely to be avalanche days.


At smaller values of FAR the POD shows greater rate of increase. Rate of POD increase for FAR values more than 0.5 is low. Avalanches depend on complex situations not represented by training data therefore the model is not confident when avalanches occur due to such reasons. Simpler situations e.g. high snowfall and snow height can be easily captured by model at lower FAR. Increasing FAR beyond a threshold gives lower gains because the threat from simpler avalanche situations

( detected at lower FAR ) can be predicted with high confidence but complex situations may not be captured and model misses such avalanche days.

We use statistics from Table 5 and Table 4 to find an acceptable FAR vs POD trade-off for operational forecasting. For example FAR of 0.3 and POD of 0.7 implies that of 52 non avalanche days approximately 15 were misclassified as

avalanche days and of 29 avalanche days approximately 21 were classified correctly [Refer to Table 6]. Therefore the model detected 21/30 avalanche days in entire season by warning for 36 days. Table 6 gives such operational performance metrics of the model for various values of FAR.


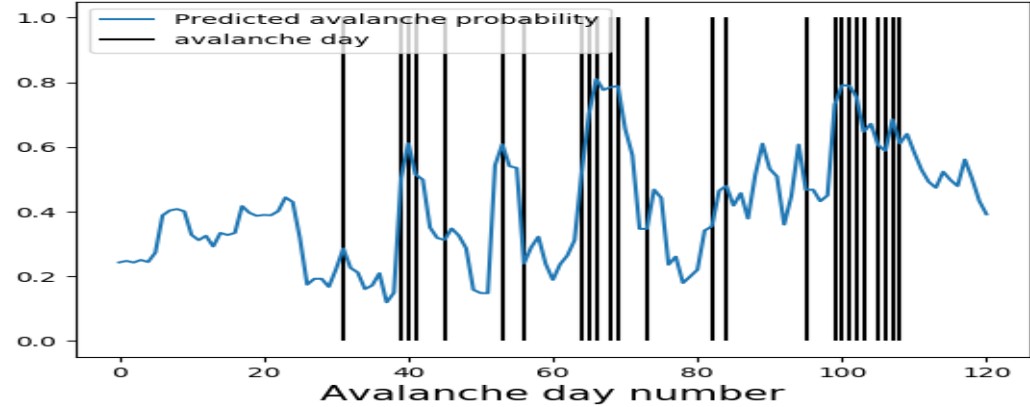


**Figure 7: Validation of model against avalanche activity in BG axis from December 2013 to March 2014. Vertical line represents an avalanche day. Blue curve shows the model predicted avalanche probability.**




**Table 6: Operational forecasting metrics of Random Forest model for BG axis.**

| FAR | Mean number of avalanche days detected out of 29 | Mean number of warnings given |
|---|---|---|
| 0.2 | 18 / 29 | 28 |
| 0.3 | 21/29 | 36 |
| 0.4 | 22/29 | 42 |
| 0.5 | 25/29 | 51 |


**4.4 Comparisons with similar models**

We compared the model with similar models based on its skill scores, selection of features, data efficiency and descriptive forecasting[ Table 7 ]. The model uses lesser data, gives informative descriptive forecasts and acceptable skill. Sufficient historical avalanche data is not available for most places therefore a data efficient model is required for forecasting.


**Table 7: Model comparisons with Random Forest model. HSS scores depend strongly on the training and testing datasets used.**

| Modeling Technique used | Highest HSS (TSS) score achieved | Training Data Used | Use of features measured which cannot be automatically measured currently | Descriptive Forecasting. |
|---|---|---|---|---|
| Support Vector Machines (Pozdnoukhov et al., 2008) | HSS = 0.62 TSS = 0.63 | 10 years data (1991 – 2000) Lochaber region, Scotland. | Yes | Suggested as future work. By exploring support vectors of trained model. |
| Calibrated nearest neighbours (Singh et al., 2014) | HSS = 0.31 | 14 years (1999 – 2012) CT Axis. | Yes | Returns a list of similar days measured by calibrated metric. |
| Calibrated nearest neighbours (Purves et al., 2003) | TSS = 0.61 | 8 years data, 1991 – 1998, Lochaber. | Yes | Returns a list of nearest days and |



| | | region , Scotland | | their attributes, visualisations of list attributes and geo map locations of similar days. |
|---|---|---|---|---|
| Random Forest (This model). | HSS =0.41 TSS = 0.42 | 3 Years ( 2010 – 2013) BG Axis. | No | Implemented by displaying decision trees in ensemble which predict high risk output. |

Descriptive Forecast (Data mining outputs) from a model is used by forecasters to understand the causes of avalanche threat. This information is used to find high risk slopes, estimate the type and magnitude of avalanches. Descriptive forecast  given by nearest neighbours is a list of most similar days to the day being forecasted. From this list forecaster makes inferrences about important variables contributing and high risk slopes. Understanding interactions between variables is difficult using this approach since numerical data about variable combinations causing high risk is unavailable. Forecasters have to  use

only few similar days, therefore variable interactions are deduced from experience largely.

Descriptive forecast by visualising trees can give information which cannot be gained using a list of nearest neighbours. Decision trees show the critical variables for a day and the range of values of these variable which were historically related to high avalanche threat. Trees can show important interactions between variables and give useful numerical data[Refer sec

5.1]. The output of a decision tree can be interpreted as a forecasting heuristic with confidence estimates from past data [See section 5 .1 for details and examples].

**5. Descriptive Forecasting**

Descriptive forecast includes information to analyse avalanche threat. Examples of descriptive forecast from some frequently used models:

   a.   Nearest neighbours model lists similar days and their attributes (Singh and Ganju, 2008; Singh et al., 2014; Purves

380        et al. 2003).

   b.   Expert systems list applicable rules (Singh and Ganju, 2008;  Schweizer and Föhn, 1996).




  c.  Support Vector Machines can list vectors which define the maximal margin hyperplane (Pozdnoukhov et al., 2008).

Precise estimates for variable combinations and their range related to high avalanche threat are difficult to make using descriptive forecasts generated by nearest neighbours model. Our model can describe important variables and their relation to avalanche risk by visualising decision trees predicting high risk. Decision tree shows a set of conditions justifying the avalanche threat and confidence estimates from past data. For models mentioned in section 4.4 such conditions need to be infered manually, causing more subjective bias. The reasoning given by the model can help forecasters validate their assumptions about the current situation or alert them if these assumptions are invalid.


**5.1 Decision tree visualisation and results**

A path from root to a leaf node in a decision tree can be interpreted as a sequence of conditions. Forecasting rules can be
defined by these condition sequences. Descriptive forecast is generated by visualising trees predicting highest avalanche threat probability. Visualised trees give rule based forecasting logic and the strength of its predictive value. In our experiments the trees show non-trivial logic which may be difficult to discover otherwise.

We show here a sample descriptive forecast for BG axis on 1-Feb-17, a day classified as high risk with predicted avalanche probability 0.54 . Ten decision trees which predicted probabilities greater than or equal to 0.9 were visualised. Most
visualised trees show that snowfall in past 10 days and high wind speed caused risk. Tree in Figure 9 demonstrates this reasoning pattern:

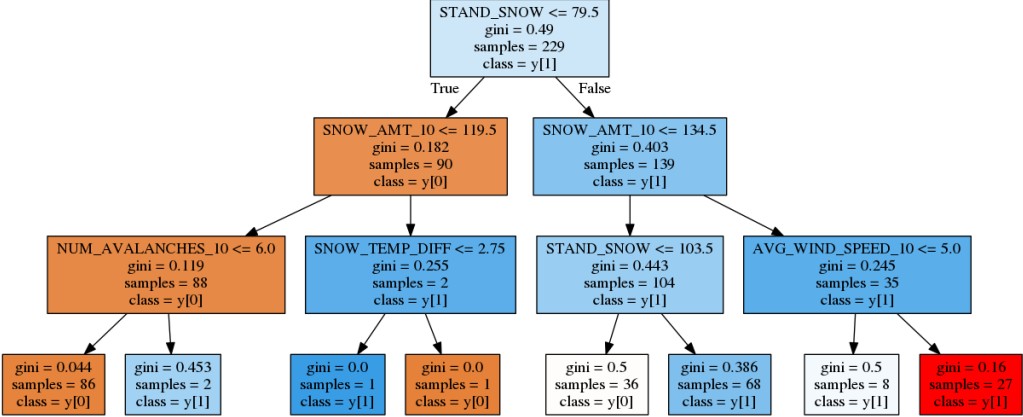

**Figure 8: Tree demonstrating risk factors on 1 – Feb – 17 at BG axis, selected from ensemble for visualisation due to its high risk output.**





Following the reasoning path from root to red leaf node we get the following heuristic satisfied for the present day:

If Standing snow > 79.5 AND Snow fall in past 10 days > 134.5 AND Average wind speed in past 10 days > 5.0 then the risk of avalanche is high ( > 90% )

Such reasoning is known to experienced forecasters. In this case model gives numerical estimates for intuition. Trees can also suggest patterns which are difficult for forecasters to observe manually. Figure 10 demonstrates such a pattern , this was visualised for descriptive forecast of 28-Mar-17.


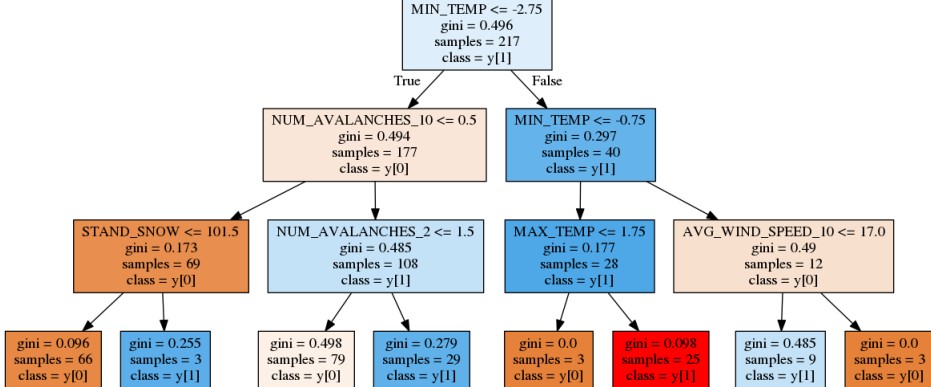

**Figure 9: Tree demonstrating risk factors on 28 – Mar – 17 at BG axis,selected for visualisation from ensemble due to its high risk output.The tree indicates that melting maybe a major reason for threat. Numerical thresholds obtained can be helpful for further data mining.**


Suggested rule satisfied for day is:

-2.75 <   MIN_TEMP <= -0.75   AND MAX_TEMP >= 1.75 then avalanche risk is high.

The temperature bounds show that snow melt maybe causing high threat. We check this hypothesis by additional data mining. Statistics from a filtered database containing only days which satisfy these bounds are compared to same statistics

from the original database[Table 8]. Other features correlated to the temperature bounds may be causing actual threat. To rule this out we made a simple univariate analysis, variables with significantly different distributions in filtered and original sets were analysed. Of these we believe only snow height is a variable leading to significant changes in hazard levels. To



analyse effect of snow height we make another filtered dataset from the original data containing only days where snow height is greater than the mean snow height of temperature filtered data. Statistics from these three datasets are compared in
Table 8.


**Table 8: Verification of decision tree output by comparing statistics of data filtered using tree output and unfiltered/control datasets.**

| Statistic | Original Dataset [ Unfiltered ] | Filtered Dataset [ by temperature bounds] | Filtered Dataset [ by snow height > 100 cm ] |
|---|---|---|---|
| Avalanche Probability on a day | 0.21 | 0.43 | 0.4 |
| Mean Standing Snow | 81 cm | 101 cm | 145 cm |
| Probability of avalanche after snow fall between 0 to 20 cm. | 0.27 | 0.61 | 0.45 |
| Probability of avalanche after snow fall between 20 to 40 cm. | 0.41 | 0.68 | 0.5 |
| Probability of avalanche after snow fall more than 40cm. | 0.51 | 0.71 | 0.58 |

Snowfall causes significantly higher risk when the temperature bounds are satisfied. In our analysis the contribution of temperature was much higher than standing snow height [see Table 8]. Higher triggering risks after snowfall at these temperatures is likely due to the formation of melt freeze crusts and higher snow density at higher temperatures. When rule is satisfied and no snowfall occurs the risk is higher than days when mean snow height is much higher , this suggests significant melting instability.


Temperature bounds can be similarly used in data base queries to get other important information from past data. Past high risk slopes which triggered under such temperature conditions, size of avalanches triggered, stability and snow profile data collected under similar temperature conditions can be searched from filtered databases.




## 6. Discussion

The model gives acceptable forecast accuracy of triggering risk by fresh snowfall or other natural causes. In 51 warnings it detected 25 out of an average of 29 avalanche days per winter [ Table 6 ]. On average half warnings of natural triggering are

true, this precision is reasonable given the difficulty of predicting natural avalanches. The false alarms can indicate untriggered snow instability. Descriptive forecast can provide more information about nature of these instabilities and their probable locations.

The example in Figure 10 gives the following rule:

-2.75 <  MIN_TEMP <= -0.75   AND MAX_TEMP >= 1.75 then avalanche risk is high( > 90% natural triggering risk).
The rule seems to predict melt avalanches, such a simple yet effective rule in terms of temperature only is difficult to find for a forecaster.  The data mining results in table 8 show that snowfall when the rule is satisfied leads to higher triggering probability. This is due to combination of factors: formation of melt freeze crusts and higher density of fresh snow at higher temperatures. The fresh snow bonds poorly with crust , due to its higher density it is also more likely to slip from crust.

When rule is satisfied and no snowfall occurs the risk is higher than days when mean snow height is much higher , this suggests significant melting instability.

Such complicated reasoning was accounted by model without any significant feature engineering effort. An explanation of data effeciency is that decision trees model such reasoning and ensemble accounts for the different causes of avalanches.

Variables involved for avalanche threat are  different for various situations therefore in avalanche datasets the important variables involved in causing threat vary across the sample space. Nearest neighbour models are unable to adopt to this variation in feature importance, they use the same distance metric to forecast in every neighbourhood of sample space. Trees in ensemble consider different features important hence this method can account for the differences in important variables. The trees trained with splitting features matching the important features for input day give higher probabilty outputs than

other trees.

Prediction is made using only parameters which can be measured automatically.  Therefore such models can use data from dense sensor grid to improve performance.  If additional parameters are required to improve forecasting process, less record

of these new parameters is required for training an updated model. Therefore data effeciency of a model implies that economic returns from setting up and updating a sensor grid can be obtained in a reasonable time period. We expect the following approaches are promising for improvement of data effeciency:

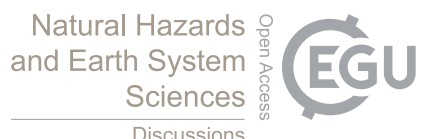

1. Use of transfer learning techniques to use data from several regions to forecast in a region where long avalanche data records are unavailable.

2. Inclusion of numerical snowpack simulation data into input features.

3. Tuning of algorithm for avalanche forecasting, changing the bagging and feature splitting procedures to account for differing importance of various situations in forecasting.


## 7. Conclusions

Requirement of long term training data is a significant problem in operational use of machine learning models for avalanche
forecasting. Data efficiency can reduce the cost of training a new model for a location or retraining an existing model to use different data. This paper demonstrates the use of Random Forest technique for avalanche forecasting on a dataset from BG-axis. The model shows significantly higher data effeciency than current operational models surveyed. This is likely due to the ability of decision trees to model specific avalanche forecast knowledge and the ensemble modeling the stochastic properties of data. The model gives acceptable forecast skill while using low amount of training data [3 winters data]. Future
work can explore reducing the data requirements further by using transfer learning techniques and specialised tuning for avalanche forecasting.

Data used by model for prediction on a day can be collected automatically, forecasts can be generated automatically. Automated data collected in high volume from a dense sensor grid can be used for generating forecasts. Data efficiency and
automated forecasting make the model economical for operational forecasting aplications.

Descriptive forecasting by visualising decision trees can give reasons for avalanche threat and help forecasters judgement by giving them numerical estimates and qualitative analysis of situation. Variable combinations causing threat and risk probabilities given the variable ranges are clear from decision trees. Further data mining can be done using these ranges and
variables to find high risk slopes and type of instabilities.



**7. Code Availability**

Not permitted


**8. Data Availability**

Not permitted

**9. Sample Availability**

Not applicable to our work

**10. Video Supplement**

None

11. **Appendices**

None, Tables in text are attached after referrences.

**12. Author contributions**

Manesh Chawla:

Implementation and validation of model.

Data exploration and visualisation.

Writing manuscript.


Amreek Singh:

Help with writing Section 2 and 3.

Suggestions for model improvement.

Model performance analysis



Review of manuscript.

## 13. Competing Interests

Authors were employed by Snow and Avalanche Study Establishment, a lab of Defence Research and Development
organisation.

## 14. Disclaimer

The information contained in this paper is true and complete to the best of our knowledge. The authors disclaim any liability
in connection with the use of this information.


## 15. Acknowledgements

We are thankful to field workers of Snow and Avalanche Study Establishment for collecting avalanche data and maintaining
the field equipment in difficult weather conditions.
The model code was written using sci-kit learn python library, all graphs were made using matplotlib.

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

**Tables**

Table 1: Parameters used for prediction. These can be automatically recorded.

| Parameter name | Unit | Description |
|---|---|---|
| MAX-TEMP | Degree Celcius ( ℃ ) | Maximum Temperature of past 24 hours |
| MIN-TEMP | Degree Celcius ( ℃ ) | Minimum Temperature of past 24 hours |
| SNOW-TEMP | Degree Celcius ( ℃ ) | Snow surface temperature measured at 0830 hours |
| SNOW-DEPTH | Meter ( m ) | Height of snow surface above ground level |
| SNOW-AMT | Meter ( m ) | Amount of snowfall in past 24 hours |
| AVG-WIND-SPEED | Meter/Second ( $m/s^{-1}$ ) | Average wind speed in past 24 hours |










**Table 2: Parameters derived from observed parameters. *Represents parameters derived using training labels.**

| Parameter name | Unit | Description |
|---|---|---|
| SNOW-TEMP-DIFF | Degree Celcius ( ºC ) | Snow surface temperature difference from past day. |
| SNOW-AMT-2 | Meter (m) | Snow fall in past 2 days. |
| SNOW-AMT-4 | Meter (m) | Snow fall in past 4 days. |
| SNOW-AMT-10 | Meter (m) | Snow fall in past 10 days. |
| NUM-AVALANCHES-2* | None | Number of avalanches in past 2 days |
| NUM-AVALANCHES-4* | None | Number of avalanches in past 4 days |
| AVG-WIND-SPEED-2 | Meter/Second ( m/s$^{-1}$ ) | Average wind speed of past 2 days |
| AVG-WIND-SPEED-4 | Meter/Second ( m/s$^{-1}$ ) | Average wind speed of past 4 days |
| AVG-WIND-SPEED-10 | Meter/Second ( m/s$^{-1}$ ) | Average wind speed of past 10 days |





**Table 3: Performance measures used for model validation.**

| Measure name | Description | Expression in terms of confusion matrix |
|---|---|---|
| False Alarm Rate(FAR) | Conditional probability of returning an avalanche day given underlying day is non avalanche day. | $\dfrac{a_{10}}{\sum a_{i0}}$ |
| Probability of Detection(POD) | Conditional probability of forecasting an avalanche day given underlying day is avalanche day. | $a_{ij} = \left\| S_{ij} \right\|$ |
| Precision | Fraction of predicted days which are avalanche days. | $\dfrac{a_{11}}{a_{10} + a_{11}}$ |
| Heidke Skill Score (HSS) | Measures the forecast performance of classifier over of a defined random forecast (Wilks, 1995). | $\dfrac{a_{11} a_{00} - a_{10} a_{01}}{\left(a_{11} + a_{01}\right)\left(a_{01} + a_{00}\right) + \left(a_{11} + a_{10}\right)\left(a_{10} + a_{00}\right)}$ |
| Hansen Kuipers Skill Score (TSS) | Measures the forecast performance of classifier over of a defined random forecast (Wilks, 1995). | $S_{ij}$ |








**Table 4 : POD and Precision values for different FAR tolerated when using Random Forest model for forecasting.**

| FAR | POD | PRECISION |
|---|---|---|
| 0.1 | 0.43 | 0.59 |
| 0.15 | 0.5 | 0.52 |
| 0.2 | 0.62 | 0.51 |
| 0.25 | 0.66 | 0.47 |
| 0.3 | 0.7 | 0.44 |
| 0.35 | 0.74 | 0.41 |
| 0.4 | 0.77 | 0.39 |
| 0.45 | 0.82 | 0.38 |
| 0.5 | 0.86 | 0.37 |
| 0.55 | 0.86 | 0.35 |
| 0.6 | 0.87 | 0.33 |
| 0.65 | 0.87 | 0.31 |
| 0.7 | 0.9 | 0.30 |






**Table 5: Summary statistics of BG axis avalanche dataset ( Dec 2010 – Mar 2017)**

| | |
|---|---|
| Number of days in winter season | 120 |
| Mean number of avalanches per season | 30 |
| Mean number of avalanches per season when snow height was greater than 50 cm. | 29 |
| Mean number of days out of 120 days per season when snow height is greater than 50 cm. | 81 |
| Mean number of non-avalanche days when snow height was greater than 50 cm. | 52 |








**Table 6: Operational forecasting metrics of Random Forest model for BG axis.**

| FAR | Mean number of avalanche days detected out of 29 | Mean number of warnings given |
|---|---|---|
| 0.2 | 18 / 29 | 28 |
| 0.3 | 21/29 | 36 |
| 0.4 | 22/29 | 42 |
| 0.5 | 25/29 | 51 |









**Table 7: Model comparisons with Random Forest model. HSS scores depend strongly on the training and testing datasets used.**

| Modeling Technique used | Highest HSS (TSS) score achieved | Training Data Used | Use of features measured which cannot be automatically measured currently | Descriptive Forecasting. |
|---|---|---|---|---|
| Support Vector Machines (Pozdnoukhov et al., 2008) | HSS = 0.62 TSS = 0.63 | 10 years data (1991 – 2000) Lochaber region, Scotland. | Yes | Suggested as future work. By exploring support vectors of trained model. |
| Calibrated nearest neighbours (Singh et al., 2014) | HSS = 0.31 | 14 years (1999 – 2012) CT Axis. | Yes | Returns a list of similar days measured by calibrated metric. |
| Calibrated nearest neighbours (Purves et al., 2003]) | TSS = 0.61 | 8 years data, 1991 – 1998, Lochaber. region , Scotland | Yes | Returns a list of nearest days and their attributes, visualisations of list attributes and geo map locations of similar days. |
| Random Forest (This model). | HSS =0.41 | 3 Years ( 2010 – | No | Implemented by |

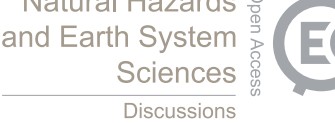

| | TSS = 0.42 | 2013) BG Axis. | | displaying decision trees in ensemble which predict high risk output. |
|---|---|---|---|---|



**Table 8: Verification of decision tree output by comparing statistics of data filtered using tree output and unfiltered/control**
**datasets.**

| Statistic | Original Dataset [ Unfiltered ] | Filtered Dataset [ by temperature bounds] | Filtered Dataset [ by snow height > 100 cm ] |
|---|---|---|---|
| Avalanche Probability on a day | 0.21 | 0.43 | 0.4 |
| Mean Standing Snow | 81 cm | 101 cm | 145 cm |
| Probability of avalanche after snow fall between 0 to 20 cm. | 0.27 | 0.61 | 0.45 |
| Probability of avalanche after snow fall between 20 to 40 cm. | 0.41 | 0.68 | 0.5 |
| Probability of avalanche after snow fall more than 40cm. | 0.51 | 0.71 | 0.58 |