# Peer review of "Data efficient Random Forest model for avalanche forecasting"

_Natural Hazards and Earth System Sciences, 2019_

## Referee Comment (RC1) · Bret Shandro (Referee) · 28 Jan 2020

**NHESS-2019-379 - Referee Comment**

January 24, 2020
Referee: Bret Shandro

**General Comments**

The scientific significance of this manuscript is good. The manuscript presents a risk management method for monitoring snow avalanches for a Himalayan highway and outlines a novel approach intended to streamline the avalanche hazard forecasting process. The methodology and conclusions make a convincing argument for potentially improving the data management process for avalanche forecasters. Specifically, for the efficient filtering of data for the avalanche forecasting process. Additionally, I agree with the authors that the false alarm rates can indicate avalanche hazard even without observed avalanches (Haegeli et al., 2010).

However, the scientific quality of this manuscript is fair because of the limited description of the weather inputs avalanche observation dataset used for training the random forest model. The manuscript communicates the study content, however, the paper may benefit from increased engagement after a thorough English language and grammatical review. Additionally, the presentation quality of the manuscript is poor.

While the technical and English language revisions required are relatively minor, the manuscript requires additional editing before publication. Figure numbers appear to be incorrect (figure 4, line 296, 323, 400, 413), figures are not labelled or explained, additional space on line 251, 267.

**Specific Comments**

The methodology could be more relevant to avalanche processes if the methodology used internationally accepted parameters for weather and snow observations (CAA, 2016).

The manuscript provides a methodology that aligns with practitioner experience for certain avalanche problems (Haegeli, Atkins, and Klassen 2010).

Line 31: Avalanche forecasting aims to identify the location of snowpack weakness and its triggering risk specific avalanche problems, their spatial distribution and sensitivity to triggering (Statham et al 2018).

Line 35: Citation required for the snowpack spatial variability, suggest Gaume et al (2014):

In numerous lines, the manuscript uses the terms 'threat', 'hazard', and 'risk' interchangeably. Avalanche hazard is a source of potential harm or loss. The potential for an avalanche(s) to cause damage to something of value. It is a function of the likelihood of triggering or frequency, and the avalanche size or magnitude. Avalanche risk is the probability of harm or cost resulting from the interaction between avalanche hazard and a specific element(s) at risk (CAA 2015, CSA 2015, Statham, 2008).

Line 80: (Pozdnoukhov et al., 2008) Pozdnoukhov et al. (2008) use support vector machines (SVM) on high dimensional feature vectors…

Section 3, the description of the study location includes the BG road axis. I believe BG road alignment would be more descriptive.

Since the model's ability is limited by the input parameters, could the input parameters derived from observed parameters be formatted like the observations gathered by avalanche practitioners? If the derived input parameters more closely match the actual forecasting process, the random forest rules may result in more insightful forecasting thresholds. For example, instead of inputting the snowfall amount in the past 2,4, and 10 days, CAA (2016) snow observations include, 12-hour and 24-hour snowfall and precipitation amounts. Another important forecasting input is the Height of Storm (HST) amount.

The authors have extensive references for relevant random forest studies. However, the manuscript does not include any references for studying avalanche observations. Weighting the avalanche days based on the number of observations or avalanche size may improve the dataset filtering and model output (Thumlert et al 2014, Laternser & Schneebeli 2003, Hägeli & McClung 2003).

Line 232: …sufficient standing snow Height of Snow (HS; CAA, 2016).

Line 231, Table 4: To address sampling bias due to a portion of the dataset having no avalanche hazard, the manuscript filters for observations with a SNOW-DEPTH > 50cm. To evaluate the validity of this assumption the reader requires the elevation of the SNOW-DEPTH parameter.

Figure 5-6 and Table 5 appear to present the same information, either the figures or the table could be removed.

The impact of the example tree diagrams could be improved by explicitly describing the colour scheme. I find it difficult to understand the reason for colour changes in the parent/child nodes. While I assume the leaf node colour scheme is green/yellow/red for low/medium/high outputs, an explicit explanation would be beneficial.

Line 425: The suggested rule only makes sense if the temperature trend is warming, however, if the trend is cooling and satisfying the rule the snowpack would likely be stabilizing. A parameter indicating temperature trend would likely assist the model with identifying useful forecasting rules.

Line 441, Table 8: The sample size should be included.

On line 466, I assume that the manuscript is describing a persistent slab avalanche problem (Heageli et al 2010) or old snow problem (Harveyet al 2009).

Table 3: The formulas presented for the confusion matrix and performance measures appear to be applied correctly. However, there is inconsistent use of a and y for different performance measures.

Figure numbers appear to be incorrect (figure 4, line 296, 323, 400, 413)

Additional space on line 251, 267.

**Technical Corrections**

Line 120-125: The reader would benefit from an explicit explanation of the random forest terms non-leaf node, leaf node, and child node.

Line 176: The  start zone elevation  of avalanche sites in the area ranges…

Line 179: Is this sentence describing the snow climate or the meteorological climate? If it is meant to describe the snow climate, please review Haegeli and McClung (2007) and Sharma & Ganju (2000) for snow climate classifications. The continental snow climate exhibits colder temperatures, more frequent periods of clear skies and less snowfall, which produces a thinner snowpack that is conducive to the formation of depth hoar and persistent weak layers (McClung and Schaerer, 2006).

**References**

Canadian Avalanche Association. (2016). Observation Guidelines and Recording Standards for Weather, Snowpack and Avalanches: Vol. 6.1.

Canadian Avalanche Association. (2015). Technical Aspects of Snow Avalanche Risk Management - Resources and Guidelines for Avalanche Practitioners in Canada. In Canadian Avalanche Association (Campbell, C. Conger, B. Gould, B. Haegeli, P. Jamieson, B. Statham, G.

Canadian Standards Association, (CSA). (2015). Risk management — Principles and guidelines. Standards Council of Canada Published (Issue CAN/CSA-ISO 31000-10).

Gaume, J., Schweizer, J., Herwijnen, a Van, Chambon, G., Reuter, B., Eckert, N., & Naaim, M. (2014). Journal of Geophysical Research : Earth Surface Evaluation of slope stability with respect to snowpack. May, 1–17. https://doi.org/10.1002/2014JF003193.Abstract

Hägeli, P., & McClung, D. M. (2003). Avalanche characteristics of a transitional snow climate-Columbia Mountains, British Columbia, Canada. Cold Regions Science and Technology, 37, 255–276. https://doi.org/10.1016/S0165-232X(03)00069-7

Haegeli, P., & McClung, D. M. (2007). Expanding the snow-climate classification with avalanche-relevant information: Initial description of avalanche winter regimes for southwestern Canada. Journal of Glaciology, 53, 266–276. https://doi.org/10.3189/172756507782202801

Haegeli, P., Atkins, R., & Klassen, K. (2010). Auxiliary material for Decision making in avalanche terrain: a field book for winter backcountry users. Canadian Avalanche Center. https://www.avalanche.ca/planning/decision-making

Harvey, S., Schweizer, J., Rhyner, H., Nigg, P., & Hasler, B. (2009). Caution-Avalanches! Institute for Snow and Avalanche Research SLF, Davos.

Laternser, M., & Schneebeli, M. (2003). Long-term snow climate trends of the Swiss Alps (1931-99). International Journal of Climatology, 23, 733–750. https://doi.org/10.1002/joc.912

McClung, D. M., & Schaerer, P. (2006). The Avalanche Handbook (3rd ed.). Mountaineers Books. https://doi.org/10.1016/0148-9062(77)90015-8

Sharma, S., & Ganju, A. (2000). Complexities of avalanche forecasting in Western Himalaya — an overview. Cold Regions Science and Technology, 31(2), 95–102. https://doi.org/10.1016/S0165-232X(99)00034-8

Statham, G., Haegeli, P., Greene, E., Birkeland, K., Israelson, C., Tremper, B., Stethem, C., McMahon, B., White, B., & Kelly, J. (2018). A conceptual model of avalanche hazard. Natural Hazards, 90(2), 663–691. https://doi.org/10.1007/s11069-017-3070-5

Statham, G. (2008). Avalanche hazard, danger and risk - a practical explanation. International Snow Science Workshop ABSTRACT, 224–227. https://doi.org/10.1016/j.coldregions.2007.09.004

Thumlert, S., Bellaire, S., & Jamieson, B. (2014). Relating avalanches to large-scale ocean–atmospheric oscillations. In International Snow Science Workshop (pp. 481–485).

---

## Referee Comment (RC2) · Anonymous Referee #2 · 5 Feb 2020

(1) This paper predicts snow avalanche using the Random Forest model. The research area is young and the paper is interesting, however, some improvements should be considered before publication:

(2) Forecast or predict? Hazard or risk? Please be consistent with using the phrases.

(3) All full names should be presented in their first occurrences, for example, CROCUS in L40, SAFRAN in L44, and etc.

(4) Literature review is missing. As a research paper, this submission needs to critically assess work previously carried out in the scientific field. Although this has been done to a limited extent in the introduction, some key references are missed. For example, Choubin et al. (2019) predicted the snow avalanche hazard using machine learning

methods.

(5) L101-102: I do not agree with using this sentence in the introduction. Transfer it to the conclusion/discussion or delete it. Instead, state the objectives clearly (preferably as listing (i) (ii), etc) at the end of the introduction.

(6) L 172: Figure 4 or Fig. 3?!

(7) What is 0830 hours?

(8) Table 2: It is mentioned that star represents parameters derived using training labels. What does that mean? Please clarify more.

(9) Table 3: Add reference for FAR, POD, and precision, too.

(10) How did you split the dataset into training and testing data? What ratio has used?

(11) The figures' number must be checked.

(12) Location of the avalanches should be presented.

(13) Tables 5 and 6 are confusing. These metrics are calculated after modeling run, but I can not see the modeling conditions in each row of the tables. There is some missing information in these Tables.

(14) Table 7: Performance of the Random Forest model is lower. So, how did you suggest this model?

Reference: Choubin, B., Borji, M., Mosavi, A., Sajedi-Hosseini, F., Singh, V.P. and Shamshirband, S., 2019. Snow avalanche hazard prediction using machine learning methods. Journal of Hydrology, 577, p.123929.

---

## Author Comment (AC1) · 19 Feb 2020

We are grateful for the careful attention, the manuscript has been revised according to the comments suggested here:

Responses to specific comments:

Line 31: Revised. Line 35: Revised( gaume et al. added)

Use of words risk, hazard done according to definitions.

Line 80: Revised ( Citations at the start of any sentence are formatted ). Section 3: We will soon add this information in a revised manuscript. Line 232: Citation added. Line 231: ????

[Figure]

Figure 5‐6 and Table 5 appear to present the same information, either the figures or the table could be removed: True, but the numerical values from Table 5 have been used to find values in Table 6 to demonstrate operational performance.

Line 425: Example was selected in march month , temperature shows increasing trend in this month mostly. We will add a trend parameter in future work. Line 441: Sample size given for each probability estimate derived in Table 8.

Table 3: Formulas corrected. Figure Numbers have been corrected.

We have cited avalanche climate literature in discussion section, following citations were added: (Rewritten discussion section added in supplements [discussion.pdf])

Hägeli, P., McClung, D. M. : Avalanche characteristics of a transitional snow climate‐Columbia Mountains, British Columbia, Canada. Cold Regions Science and Technology, 37, pp. 255–276, https://doi.org/10.1016/S0165-232X(03)00069-7 , 2003.

Haegeli, P., McClung, D. M.: Expanding the snow‐climate classification with avalanche‐relevant information: Initial description of avalanche winter regimes for southwestern Canada. Journal of Glaciology, 53, 266–276. https://doi.org/10.3189/172756507782202801, 2007

Mock C.J., Birkeland K.W. : Snow Avalanche Climatology of the Western United States Mountain Ranges, Bulletin of the American Meteoroligical Society , https://doi.org/10.1175/1520-0477(2000)081<2367:SACOTW>2.3.CO;2, 2000.

Shandro , B., Haegeli, P.: Characterizing the nature and variability of avalanche hazard in western Canada, Natural Hazards and Earth System Sciences, 18, pp. 1141–1158, https://doi.org/10.5194/nhess-18-1141-2018 , 2018.

Sharma, S.S., Ganju, A.: Complexities of avalanche forecasting in Western Himalaya — an overview, Cold Regions Science and Technology, 31(2), pp. 95-102, https://doi.org/10.1016/S0165-232X(99)00034-8 , 2000.

Thumlert, S., Bellaire, S., Jamieson, B.: Relating avalanches to large‐scale ocean–atmospheric oscillations. In International Snow Science Workshop , Banff Canada, 29 September ,pp. 481–485 , 2014.

The following citations were added to literature survey in introduction: (Rewritten introduction section added in supplements [intro.pdf])

Choubin, B., Borji, M., Mosavi, A., Sajedi-Hosseini, F., Singh, V.P., Shamshirband, S.: Snow avalanche hazard prediction using machine learning methods. Journal of Hydrology, 577, p.123929, https://doi.org/10.1016/j.jhydrol.2019.123929 ,2019.

Davis, R.E., Elder, K., Howlett, D. and Bouzaglou, E.. 1999. Relating storm and weather factors to dry slab avalanche activity at Alta, Utah, and Mammoth Mountain, California, using classification and regression trees. Cold Reg. Sci. Technol., 30(1–3), 79–90

Gaume, J., Schweizer, J., Herwijnen, a Van, Chambon, G., Reuter, B., Eckert, N., Naaim, M.: Earth Surface Evaluation of slope stability with respect to snowpack, Journal of Geophysical Research. 119(9), pp. 1783 – 1799, https://doi.org/10.1002/2014JF003193, 2014.

Rahmati, O., Ghorbanzadeh, O., Teimurian, T., Mohammadi, F., Tiefenbacher, J.P., Falah, F., Pirasteh, S., Ngo, P.T., Bui, D.T.: Spatial Modeling of Snow Avalanche Using Machine Learning Models and Geo-Environmental Factors: Comparison of Effectiveness in Two Mountain Regions, Remote Sensing, https://doi.org/10.3390/rs11242995 , 2019.

Rubin, M.J., Camp, T., Herwijnen. A.V, Schweizer. J.: Automatically Detecting Avalanche Events in Passive Seismic Data, 11th International Conference on Machine Learning and Applications, Boca Raton , FL, USA, 12 – 15 Dec. 2012, https://doi.org/10.1109/ICMLA.2012.12, 2012.

Schirmer, M., Lehning M., Schweizer, J.,: Statistical forecasting of regional avalanche

danger using simulated snow-cover data. Journal of Glaciology, 55(193), 761 – 768, https://doi.org/10.3189/002214309790152429, 2009.

Statham, G., Haegeli, P., Greene, E., Birkeland, K., Israelson, C., Tremper, B., Stethem, C., McMahon, B., White, B., Kelly, J: A conceptual model of avalanche hazard. Natural Hazards, 90(2), 663–691, https://doi.org/10.1007/s11069-017-3070-5 , 2018.

Thuring, T., Schoch, M., Herwijnen, A.V., Schweizer, J.: Robust snow avalanche detection using supervised machine learning with infrasonic sensor arrays, Cold Regions Science and technology, 111, pp. 60-65, https://doi.org/10.1016/j.coldregions.2014.12.014, 2015.

Technical Corrections Line 120‐125: The reader would benefit from an explicit explanation of the random forest terms non‐leaf node, leaf node, and child node. Done [ The supplement intro.pdf ends with this explanation ]

Line 176: Corrected.

Line 179: Is this sentence describing the snow climate or the meteorological climate? If it is meant to describe the snow climate, please review Haegeli and McClung (2007) and Sharma & Ganju (2000) for snow climate classifications. The continental snow climate exhibits colder temperatures, more frequent periods of clear skies and less snowfall, which produces a thinner snowpack that is conducive to the formation of depth hoar and persistent weak layers (McClung and Schaerer, 2006).

The region has continental snow climate classification according to Mock and Birkeland scheme. Additional data has been visualised in figure 3b [supplement attached as climate.png ] to give an intuition about the regions climatology. The figure gives monthwise median values for precipitation , temperature and sunshine duration of approximately 25 years ( Nov-1993 to April-2017 )
* * *
**Data efficient Random Forest model for avalanche forecasting**

Manesh Chawla[1], Amreek Singh[2]

[1]Snow and Avalanche Study Establishment, Manali - 175103, India
[2]Snow and Avalanche Study Establishment, Chandigarh - 160037, India

5 *Correspondence to*: Manesh Chawla (zmfzmj123@gmail.com)

**Abstract.**

Fast downslope release of snow (avalanche) is a serious hazard to people living in snow bound mountains. Released snow mass can gain sufficient momentum on its down slope path to kill humans, uproot trees and rocks, destroy buildings. Direct
10   reduction of avalanche threat is done by building control structures to add mechanical support to snowpack and reduce or deflect downward avalanche flow. On large terrains it is economically infeasible to use these methods on each hazard site. Therefore forecasting and avoiding avalanches is the only feasible method to reduce hazard, but sufficient snow stability data for accurate forecasting is generally unavailable and difficult to collect. Forecasters infer snow stability from their knowledge of local weather, terrain and sparsely available snowpack observations. This inference process is vulnerable to
15   human bias therefore machine learning models are used to find patterns from past data and generate helpful outputs to minimise and quantify uncertainty in forecasting process. These machine learning techniques require long past records of avalanches which are difficult to obtain. In this paper we propose a data efficient Random Forest model to address this problem. The model can generate a descriptive forecast showing reasoning and patterns which are difficult to observe manually. Our model advances the field by being inexpensive and convenient for operational forecasting due to its data
20   efficiency, amenable to automation and ability to describe its decisions.

**1 Introduction**

In snow bound areas avalanches cause loss of life and property worldwide. Avalanche deaths are estimated at 250 per year (Schweizer et al., 2015). Government and private agencies are funded to reduce avalanche risk for important activities and property e.g. road/rail transport, construction, border patrolling etc. This effort has led to development of several
25   techniques to reduce avalanche risk. Avalanche hazard mapping is done to estimate the long term hazard at each slope in a region (Choubin et al., 2019; Rahmati et al., 2019). The map is used to plan active risk reduction methods e.g. building control structures, modification of nearby terrain or use of explosives to trigger avalanches in controlled way (Fuchs et al., 2007). Using active techniques at each hazard site is economically infeasible therefore avalanche forecasting is practised to reduce avalanche exposure. Individuals can use information in forecast to minimise short term risk in snow bound areas.
30   Avalanche forecasting aims to identify the locations of snowpack weakness, their spatial distribution and sensitivity to triggering (Statham et. al. 2018). Observing snowpack stability at a high spatio-temporal resolution over large terrain is a difficult problem. Therefore stability at most risk sites is deduced using secondary observable data e.g. meteorological and snowpack parameters from a similar representative site, terrain parameters of the site, expected changes to snowpack by imminent weather etc. Snow stability shows high variance with respect to terrain features (Gaume et al., 2014). Deduction
35   process for snow stability from secondary data has not been mathematically formulated therefore forecasters need to rely on their intuition of local terrain and snowpack patterns to estimate stability and collect more information to minimise uncertainty (LaChapelle, 1980; Schweizer et al., 2008; McClung and Schaerer, 2006). Numerical and statistical models are important tools for adding objectivity to this process.

**Fig. 1.** Introduction rewritten

**285** **6. Discussion**

The model gives acceptable forecast accuracy of triggering probability by fresh snowfall or other natural causes. In 51 warnings it detected 25 out of an average of 29 avalanche days per winter [Table 6]. On average half of total warnings of natural triggering are true. This precision is reasonable given the difficulty of forecasting natural avalanches. The false alarms can indicate un-triggered snow instability. Descriptive forecast can provide more information about nature of these

**290** instabilities and their probable locations.

Consider the rule of figure 9 example:

-2.75 <  MIN_TEMP <= -0.75   AND MAX_TEMP >= 1.75 then avalanche probability = high (> 90% natural triggering probability).

This rule seems to predict melt avalanches. Such a simple yet effective rule in terms of temperature only is difficult to find

**295** for a forecaster. We checked this hypothesis by additional data mining.  Statistics from a filtered database containing only days which satisfy these bounds are compared to same statistics from the original unfiltered database [Table 8]. Other features correlated to the temperature bounds may be causing hazard. To rule this out we made a simple univariate analysis, where variables with significantly different distributions in filtered and original sets were analysed. Of these we believe only snow height is a variable leading to significant changes in hazard levels. To analyse effect of snow height, we applied

**300** another filtering to get data where snow height is greater than the mean snow height of temperature filtered data. Statistics from these three datasets are compared in Table 8.

The data mining results in Table 8 show that snowfall when the rule is satisfied leads to higher triggering probability. This is due to combination of factors: formation of melt freeze crusts and higher density of fresh snow at higher temperatures (Statham et al., 2014; Meløysund et al., 2007 ). The fresh snow bonds poorly with crust and due to its higher density it is

**305** also more likely to slip from crust. When rule is satisfied and no snowfall occurs, the triggering probability is higher than days when mean snow height is much higher. This suggests significant melting instability.

The model inferred the effect of a critical snowpack structure (melt freeze crust) from meteorological data. Capturing more complex properties of these structures e.g. persistence and strength require further feature engineering. Effect of persistent snowpack structures and climatic oscillations on avalanche activity has been analysed in detail by many researchers

**310** (Laternser and Schneebeli, 2003; Hägeli and McClung, 2003; Thumlert., et al. 2014). The resulting characterisations of avalanche climates can be used to derive relevant indexes to forecast (Haegeli and  McClung, 2007; Shandro and Haegeli, 2018). Example above demonstrates that the model can be expected to account for these complex effects using simple and relevant extracted features.

An explanation of data efficiency is that decision trees model such reasoning and ensemble accounts for the different

**315** causes of avalanches. Variables involved for avalanche hazard are different for various situations therefore in avalanche datasets the important variables involved in causing hazard vary across the sample space. Nearest neighbour models are unable to adapt to this variation in feature importance as they use the same distance metric to forecast in every neighbourhood of sample space. Trees in ensemble accord importance to different features hence this method can account for the differences in important variables. The trees trained with splitting features matching the important features for input

**320** day give higher probability outputs than other trees.

Prediction is made using parameters which can be measured automatically too. Therefore such models can use data from dense sensor grid to improve performance. If additional parameters are required to improve forecasting process, only a few records of these new parameters are required for training an updated model. Therefore data efficiency of a model implies

14

**Fig. 2.** discussion section rewritten

**Fig. 3.** climate figure

---

## Author Comment (AC2) · 19 Feb 2020

We are grateful for your careful attention to manuscript and suggestions for improving it. We have fixed the issues suggested. Editing quality of manuscript has been improved.

Our pointwise response is listed below:

(2) This issue has been fixed. We use hazard to refer the damage potential of avalanche, risk measures the chance of damage to an exposed object.

(3) We could not find full names of some models in their original papers. [SNOWPACK , CROCUS ].

(4) We cited the following additional papers using Machine Learning to do Avalanche

[Figure]

Hazard Mapping, avalanche detection, forecasting, snow density prediction from meteorological variables:

Choubin, B., Borji, M., Mosavi, A., Sajedi-Hosseini, F., Singh, V.P., Shamshirband, S.: Snow avalanche hazard prediction using machine learning methods. Journal of Hydrology, 577, p.123929, https://doi.org/10.1016/j.jhydrol.2019.123929 ,2019.

Davis, R.E., Elder, K., Howlett, D. and Bouzaglou, E.. 1999. Relating storm and weather factors to dry slab avalanche activity at Alta, Utah, and Mammoth Mountain, California, using classification and regression trees. Cold Reg. Sci. Technol., 30(1–3), 79–90

Meløysund, V., Leira, B., Høiseth, K.V., Lisø, K.R.: Predicting snow density using meteorological data. Meteorological Observations, 14, 413 – 423, https://doi.org/10.1002/met.40, 2007.

Rahmati, O., Ghorbanzadeh, O., Teimurian, T., Mohammadi, F., Tiefenbacher, J.P., Falah, F., Pirasteh, S., Ngo, P.T., Bui, D.T.: Spatial Modeling of Snow Avalanche Using Machine Learning Models and Geo-Environmental Factors: Comparison of Effectiveness in Two Mountain Regions, Remote Sensing, https://doi.org/10.3390/rs11242995 , 2019.

Rubin, M.J., Camp, T., Herwijnen. A.V., Schweizer. J.: Automatically Detecting Avalanche Events in Passive Seismic Data, 11th International Conference on Machine Learning and Applications, Boca Raton , FL, USA, 12 – 15 Dec. 2012, https://doi.org/10.1109/ICMLA.2012.12, 2012.

Schirmer, M., Lehning M., Schweizer, J.,: Statistical forecasting of regional avalanche danger using simulated snow-cover data. Journal of Glaciology, 55(193), 761 – 768, https://doi.org/10.3189/002214309790152429, 2009.

Thuring, T., Schoch, M., Herwijnen, A.V., Schweizer, J.: Robust snow avalanche detection using supervised machine learning with infrasonic

sensor arrays, Cold Regions Science and technology, 111, pp. 60-65, https://doi.org/10.1016/j.coldregions.2014.12.014, 2015.

Additional literature about avalanche climatology, avalanche forecasting, snow stability evaluation has been cited: [in discussion section , revised section is atteched as discussion.pdf ]

Hägeli, P., McClung, D. M. : Avalanche characteristics of a transitional snow climate‐Columbia Mountains, British Columbia, Canada. Cold Regions Science and Technology, 37, pp. 255–276, https://doi.org/10.1016/S0165-232X(03)00069-7 , 2003.

Haegeli, P., McClung, D. M.: Expanding the snow‐climate classification with avalanche‐relevant information: Initial description of avalanche winter regimes for southwestern Canada. Journal of Glaciology, 53, 266–276. https://doi.org/10.3189/172756507782202801, 2007

Mock C.J., Birkeland K.W. : Snow Avalanche Climatology of the Western United States Mountain Ranges, Bulletin of the American Meteoriligical Society , https://doi.org/10.1175/1520-0477(2000)081<2367:SACOTW>2.3.CO;2, 2000.

Shandro , B., Haegeli, P.: Characterizing the nature and variability of avalanche hazard in western Canada, Natural Hazards and Earth System Sciences, 18, pp. 1141–1158, https://doi.org/10.5194/nhess-18-1141-2018 , 2018.

Sharma, S.S., Ganju, A.: Complexities of avalanche forecasting in Western Himalaya — an overview, Cold Regions Science and Technology, 31(2), pp. 95-102, https://doi.org/10.1016/S0165-232X(99)00034-8 , 2000.

Thumlert, S., Bellaire, S., Jamieson, B.: Relating avalanches to large‐scale ocean–atmospheric oscillations. In International Snow Science Workshop , Banff Canada, 29 September ,pp. 481–485 , 2014.

(5) Stated as objectives. [ introduction rewritten and attached as intro.pdf ]

(6) All figure numbers have been corrected.

(7) The time of observation of SNOW_TEMP(Snow Surface Temperature) is 8:30 AM(Morning) , mentioned as 0830 hours here. We have given time as 8:30AM.

(8) Rewritten that it is derived using past avalanche activity , so there is no confusion.

(9) Cited Wilks.

(10) This is written in the first sentence of section 4.1. We can write this in captions of tables/graphs if it improves readability.

(11) All figure numbers have been corrected.

(12) The area is large with several avalanche sites, it may be difficult to visualize them in this paper.

(13) We have summarised the modeling conditions and details of evaluation procedure at the start of section 4.2.

(14) The performance is acceptable for operational forecast [ Table 6 ], model is recommended due to this performance with low amount of data and its ease of automation. Performance can be improved with lesser data, this is an area for future research: we have built a transfer learning model prototype with better performance and data effeciency. Adding other simulated snowpack data can also bring improvements.
* * *
**Data efficient Random Forest model for avalanche forecasting**

Manesh Chawla[1], Amreek Singh[2]

[1]Snow and Avalanche Study Establishment, Manali - 175103, India
[2]Snow and Avalanche Study Establishment, Chandigarh - 160037, India

5  *Correspondence to*: Manesh Chawla (zmfzmj123@gmail.com)

**Abstract.**

Fast downslope release of snow (avalanche) is a serious hazard to people living in snow bound mountains. Released snow mass can gain sufficient momentum on its down slope path to kill humans, uproot trees and rocks, destroy buildings. Direct
10  reduction of avalanche threat is done by building control structures to add mechanical support to snowpack and reduce or deflect downward avalanche flow. On large terrains it is economically infeasible to use these methods on each hazard site. Therefore forecasting and avoiding avalanches is the only feasible method to reduce hazard, but sufficient snow stability data for accurate forecasting is generally unavailable and difficult to collect. Forecasters infer snow stability from their knowledge of local weather, terrain and sparsely available snowpack observations. This inference process is vulnerable to
15  human bias therefore machine learning models are used to find patterns from past data and generate helpful outputs to minimise and quantify uncertainty in forecasting process. These machine learning techniques require long past records of avalanches which are difficult to obtain. In this paper we propose a data efficient Random Forest model to address this problem. The model can generate a descriptive forecast showing reasoning and patterns which are difficult to observe manually. Our model advances the field by being inexpensive and convenient for operational forecasting due to its data
20  efficiency, amenable to automation and ability to describe its decisions.

**1 Introduction**

In snow bound areas avalanches cause loss of life and property worldwide. Avalanche deaths are estimated at 250 per year (Schweizer et al., 2015). Government and private agencies are funded to reduce avalanche risk for important activities and property e.g. road/rail transport, construction, border patrolling etc. This effort has led to development of several
25  techniques to reduce avalanche risk. Avalanche hazard mapping is done to estimate the long term hazard at each slope in a region (Choubin et al., 2019; Rahmati et al., 2019). The map is used to plan active risk reduction methods e.g. building control structures, modification of nearby terrain or use of explosives to trigger avalanches in controlled way (Fuchs et al., 2007). Using active techniques at each hazard site is economically infeasible therefore avalanche forecasting is practised to reduce avalanche exposure.  Individuals can use information in forecast to minimise short term risk in snow bound areas.
30  Avalanche forecasting aims to identify the locations of snowpack weakness, their spatial distribution and sensitivity to triggering (Statham et. al. 2018). Observing snowpack stability at a high spatio-temporal resolution over large terrain is a difficult problem. Therefore stability at most risk sites is deduced using secondary observable data  e.g. meteorological and snowpack parameters from a similar representative site, terrain parameters of the site, expected changes to snowpack by imminent weather etc. Snow stability shows high variance with respect to terrain features (Gaume et al., 2014). Deduction
35  process for snow stability from secondary data has not been mathematically formulated therefore forecasters need to rely on their intuition of local terrain and snowpack patterns to estimate stability and collect more information to minimise uncertainty (LaChapelle, 1980; Schweizer et al., 2008; McClung and Schaerer, 2006).  Numerical and statistical models are important tools for adding objectivity to this process.

**Fig. 1.**

**6. Discussion**

The model gives acceptable forecast accuracy of triggering probability by fresh snowfall or other natural causes. In 51 warnings it detected 25 out of an average of 29 avalanche days per winter [Table 6]. On average half of total warnings of natural triggering are true. This precision is reasonable given the difficulty of forecasting natural avalanches. The false alarms can indicate un-triggered snow instability. Descriptive forecast can provide more information about nature of these instabilities and their probable locations.

Consider the rule of figure 9 example:

-2.75 < MIN_TEMP <= -0.75 AND MAX_TEMP >= 1.75 then avalanche probability = high (> 90% natural triggering probability).

This rule seems to predict melt avalanches. Such a simple yet effective rule in terms of temperature only is difficult to find for a forecaster. We checked this hypothesis by additional data mining. Statistics from a filtered database containing only days which satisfy these bounds are compared to same statistics from the original unfiltered database [Table 8]. Other features correlated to the temperature bounds may be causing hazard. To rule this out we made a simple univariate analysis, where variables with significantly different distributions in filtered and original sets were analysed. Of these we believe only snow height is a variable leading to significant changes in hazard levels. To analyse effect of snow height, we applied another filtering to get data where snow height is greater than the mean snow height of temperature filtered data. Statistics from these three datasets are compared in Table 8.

The data mining results in Table 8 show that snowfall when the rule is satisfied leads to higher triggering probability. This is due to combination of factors: formation of melt freeze crusts and higher density of fresh snow at higher temperatures (Statham et al., 2014; Meløysund et al., 2007 ). The fresh snow bonds poorly with crust and due to its higher density it is also more likely to slip from crust. When rule is satisfied and no snowfall occurs, the triggering probability is higher than days when mean snow height is much higher. This suggests significant melting instability.

The model inferred the effect of a critical snowpack structure (melt freeze crust) from meteorological data. Capturing more complex properties of these structures e.g. persistence and strength require further feature engineering. Effect of persistent snowpack structures and climatic oscillations on avalanche activity has been analysed in detail by many researchers (Laternser and Schneebeli, 2003; Hägeli and McClung, 2003; Thumlert., et al. 2014). The resulting characterisations of avalanche climates can be used to derive relevant indexes to forecast (Haegeli and McClung, 2007; Shandro and Haegeli, 2018). Example above demonstrates that the model can be expected to account for these complex effects using simple and relevant extracted features.

An explanation of data efficiency is that decision trees model such reasoning and ensemble accounts for the different causes of avalanches. Variables involved for avalanche hazard are different for various situations therefore in avalanche datasets the important variables involved in causing hazard vary across the sample space. Nearest neighbour models are unable to adapt to this variation in feature importance as they use the same distance metric to forecast in every neighbourhood of sample space. Trees in ensemble accord importance to different features hence this method can account for the differences in important variables. The trees trained with splitting features matching the important features for input day give higher probability outputs than other trees.

Prediction is made using parameters which can be measured automatically too. Therefore such models can use data from dense sensor grid to improve performance. If additional parameters are required to improve forecasting process, only a few records of these new parameters are required for training an updated model. Therefore data efficiency of a model implies

14

**Fig. 2.**

---

## Author Comment (AC3) · 20 Feb 2020

We propose to replace the exiting manuscript with revised manuscript which carries major revision as per comments of reviewers, for proper appreciation of the paper and further review/processing. Please advice.